# Development of a Frugal, In Situ Sensor Implementing a Ratiometric Method for Continuous Monitoring of Turbidity in Natural Waters

**DOI:** 10.3390/s23041897

**Published:** 2023-02-08

**Authors:** Raul Sanchez, Michel Groc, Renaud Vuillemin, Mireille Pujo-Pay, Vincent Raimbault

**Affiliations:** 1LAAS-CNRS, Université de Toulouse, CNRS, 31400 Toulouse, France; 2Sorbonne Université, CNRS, FR3724, Observatoire Océanologique de Banyuls, 66651 Banyuls-sur-Mer, France; 3Sorbonne Université, CNRS, UMR7621, Laboratoire d’Océanographie Microbienne (LOMIC), 66651 Banyuls-sur-Mer, France

**Keywords:** turbidity, frugal sensors, ratiometric, in situ, water quality

## Abstract

Turbidity is a commonly used indicator of water quality in continental and marine waters and is mostly caused by suspended and colloidal particles such as organic and inorganic particles. Many methods are available for the measurement of turbidity, ranging from the Secchi disk to infrared light-based benchtop or in situ turbidimeters as well as acoustic methods. The operational methodologies of the large majority of turbidity instruments involve the physics of light scattering and absorption by suspended particles when light is passed through a sample. As such, in the case of in situ monitoring in water bodies, the measurement of turbidity is highly influenced by external light and biofouling. Our motivation for this project is to propose an open-source, low-cost in situ turbidity sensor with a suitable sensitivity and operating range to operate in low-to-medium-turbidity natural waters. This prototype device combines two angular photodetectors and two infrared light sources with different positions, resulting in two different types of light detection, namely nephelometric (i.e., scattering) and attenuation light, according to the ISO 7027 method. The mechanical design involves 3D-printed parts by stereolithography, which are compatible with commercially available waterproof enclosures, thus ensuring easy integration for future users. An effort was made to rely on mostly off-the-shelf electronic components to encourage replication of the system, with the use of a highly integrated photometric front-end commonly used in portable photoplethysmography systems. The sensor was tested in laboratory conditions against a commercial benchtop turbidimeter with Formazin standards. The monitoring results were analyzed, obtaining a linear trendline from 0 to 50 Nephelometric Turbidity Unit (NTU) and an accuracy of +/−0.4 NTU in the 0 to 10 NTU range with a response time of less than 100 ms.

## 1. Introduction

Turbidity is an important indicator of water quality in rivers, streams, lakes, seas, and watersheds, and as such, it is a key parameter for environmental studies as well as for the health of human intake [1]. It is basically a physical property of fluids that measures the cloudiness of water and is influenced by the presence of suspended and dissolved particles that block or scatter the light in water bodies, thus modifying water transparency [2]. These particles can be of organic or inorganic origin. In the case of organic materials, high turbidity can indicate presence of microorganisms such as bacteria or events such as algae blooms. In the case of inorganic materials, high turbidity can indicate high amounts of suspended sediments such as clay or silt, which is caused by erosion [3,4]. Besides human interference, the environment’s turbidity level can be influenced by nutrient runoff or soil erosion from farming [5] but also by geological disturbances that can cause turbidity currents [6,7].

Commercial turbidity systems are mostly offline systems that require the action of a trained operator to collect the samples and perform the analysis either on site (portable system) or in a laboratory (benchtop system). While this approach generally offers the most precise turbidity measurements, it limits the spatial and temporal resolution. This punctual water sampling also does not allow to observe sudden events, and in addition to the equipment cost, there are also operational costs (human resources, travel, sample storage, etc.). In some cases, it is thus highly desirable to use in situ turbidity sensors; while commercial in situ turbidity sensors are readily available, their adoption is limited by their high cost (several thousands of U.S. dollars typically), which also limits spatial and temporal resolution. While the need for a low-cost, in situ turbidity sensor has already been explored in the literature, our goal is to back up these efforts with a sensor that can be used in low-turbidity areas such as the French coastal area of the Mediterranean Sea as well as in more turbid freshwater systems.

### 1.1. Turbidity Measurement Methods

While complementary methods such as acoustic [8] or time-resolved [9] ones are also used for specific cases, turbidity is mostly measured optically by a combination of a light source and one or more photodetectors that measure the scattering and/or absorption properties of particles suspended in the water sample. While absorption is a directional measurement, scattering occurs in all directions, with a diffraction pattern dependent on the particle size [2]; hence, different optical configurations can be implemented. The most common configurations are represented Figure 1. Depending on the angle between the light source and the detector, they are referred as nephelometric (angle = 90°), attenuation (angle = 180°), backscattering (0° < angle < 90°), or forward scattering (90° < angle < 180°). Based on the angle used for the measurements, different types of units are used, the most common being Nephelometric Turbidity Unit (NTU), but other units such as Formazin Nephelometric Unit (FNU) or Formazin Attenuation Unit (FAU) can be encountered [4].

Each configuration will behave differently regarding turbidity. Backscattering is considered to be suitable to high turbidity values only (>1000 NTU), and as such, it is not of direct interest in our work. Nephelometric (90° detection angle) is considered the best angle to measure scattered light regardless of particles size [10], but is advised to be used between 0 to 40 NTU, where light intensity and turbidity have a linear relationship [11]. Attenuation (180° detection angle) measures the transmitted light through the sample and is affected by the combined effects of scattering and absorption: an increase in turbidity translates to a decrease of transmitted light. The attenuation method is only recommended for turbidity levels over 40 NTU [5,12]; however, it is also used as a secondary detector in combination with a 90° detector in ratiometric designs. Turbidity measurements are normalized by internationally recognized certification organisms. The main approved turbidity methods are ISO 7027-1 and US-EPA Method 180.1, but other methods such as Standard Methods 2130B and Great Lakes Instrument Method 2 (GLI Method 2) are also endorsed by the US-EPA [12,13,14,15]. A good overview of the different configurations used in each method can be obtained in references [15,16].

### 1.2. Turbidity Sensors Calibration

Due to the diversity of turbidity sources, the calibration of turbidimeters using natural sediment sources is problematic in most cases, especially for intercomparing between different instruments. To obtain a more standardized, repeatable calibration method, a polymer called Formazin [17] has been adopted by most of the manufacturers in the industry. It is prepared by mixing solutions of hydrazine sulfate and hexamethylenetetramine in water [18] to obtain different chain lengths in random configurations, covering a range of particle shapes and sizes from less than 0.1 to over 10 microns, making it a relatively straightforward light-scattering calibration standard. One of its main advantages is that it can be repeatably and reproducibly prepared from raw materials into a calibrated stock solution that is diluted to obtain virtually any concentration. Under proper storage conditions, Formazin standards are stable over a year except for very low concentrations (<2 NTU), where long-term stability is degraded.

Although it is the calibration standard of choice of the most common official turbidity measurement methods, Formazin has also a couple of inherent drawbacks, which are summed up in Kitchener et al.’s work [19]. In particular, the shape of Formazin particles is not normalized although particles shapes can have strong influence on side-scattering. Is should also be stated that uncertainties arose due to the high dilution ratios typically required at low turbidity, reinforced by the lack of stability of these highly diluted solutions. It is commonly observed that when used with the same Formazin calibration solution, commercial turbidimeters that fulfil requirements of the same official standard (EPA/ISO) can give different turbidity values. This has been observed on laboratory benchtop instruments but also for in situ instruments [20,21]. Research on better calibration methods of existing turbidimeters as well as design of new instruments that overcomes the lack of comparability between current instruments [22] are out of the scope of this present work, but some design recommendations are incorporated in our sensor as suggested by other authors.

### 1.3. Commercial and Research-Level Instruments

As a ubiquitous water quality parameter, many commercial instruments are available to measure turbidity. The vast majority is based on optical measurement in the infrared, using side-scattering, back-scattering, attenuation, or a combination of these in order to satisfy the officially endorsed methods described earlier. The instruments can be classified within three categories: (i) benchtop instruments, which offer the best accuracy; (ii) hand-held portable devices, which are the least-expensive options; (iii) inline sensors, which are dedicated to analysis in water pipes; and (iv) in situ sensors, which can be “self-contained” or available as an add-on for multiparameter sondes. Both (i) and (ii) require sampling the water bodies for further analysis, and as such, they are not adapted for real-time monitoring, remote monitoring, or high spatio-temporal resolution measurements, as they would require an impractical amount of work for sampling, storage, and analysis.

Pricewise, a commercial turbidimeter costs between USD 600 to more than USD 5000, with portable hand-held devices being the most affordable option, while high-precision benchtop instruments tend to be the most expensive. In situ sensors, which are the scope of this paper, are usually in the middle of the range, but most of the time, they need additional equipment such as a logger or a display, for example, which makes a complete setup cost several hundreds of US dollars. Due to the constraints of in situ measurements of water bodies, which include instrument damages due to natural phenomenon, robbery, and degradation by humans or wildlife and the necessity in some cases to collect data at a better spatio-temporal resolution, the relatively high cost of commercial instruments has led to much research on alternative, low-cost turbidity sensors that, while compromising slightly on measurement quality, can provide valuable data at a fraction of the cost. Table 1 lists recent achievements reported in the literature. A comprehensive list of commercially available turbidimeters can be found in the Aquaref report [23] as well as in the inter-comparison study of Rymszewicz et al. [20] that focuses on in situ instruments.

## 2. Materials and Methods

The development of our in situ turbidity sensor is targeted toward the coastal waters at the Oceanological Observatory of Banyuls sur Mer (OOB), France. In this area of the Gulf of Lion in the Mediterranean Sea, turbidity levels are considered quite low, with average values ranging from 0 to 10 NTU typically during the year, which implies that the sensor must offer sufficient resolution (i.e., 0.5 NTU or better). Biofouling is also a common occurrence during long-term deployment of optical sensors in this area, as observed at the OOB as a part of the French Coastal Monitoring Network SOMLIT.

Based on these constraints, we chose to design our sensor around the Great Lakes Instruments Method 2 (GLI-2), also referred to as modulated four-beam turbidimeter, which uses two light sources (infrared LEDs) and two photodetectors (photodiodes) to perform a ratiometric measurement that combines nephelometric and attenuation readings. This method improves instrument stability by cancelling out errors due to the degradation of the light source, water color effects, or fouling on the sensor windows [19]. Even if all four optical ports are partially blocked, this method can still provide accurate turbidity measurements [30]. The LEDs alternate light pulses periodically, and the two photodetectors take simultaneous readings, providing an active signal and reference signal. This operating principle is summarized Figure 2. The operating range is typically 0–100 NTU; however, it loses some accuracy in levels above 40 NTU. GLI-2 is known to be very accurate for lower-turbidity ranges, in particular within the 0–1 NTU range [15], which makes this type of instrument desirable for water bodies with low turbidity. The capability to limit the influence of light-source drift and fouling also are a plus when considering in situ deployment. However, the design layout makes this method harder to integrate into a field deployable instrument compared to a conventional nephelometer, where both the light source and the photodetector can be protected by a flat optical surface. Another advantage of the GLI-2 design is the ability to obtain information on side-scattered light (nephelometric) and attenuation (transmission), with the latter recommended for inclusion in new turbidity instrumentation by Kitchener et al. [19], as it allows for the use of SI-based units for calibration. Compared to conventional nephelometric instruments that are calibrated with Formazin, this allows better intercomparison with other turbidimeters, a characteristic that is currently lacking from commercial systems, as highlighted by Rymszewicz et al. [20]. To our knowledge, our sensor is the first academic work on a GLI-2-based design that can operate continuously in situ.

### 2.1. Overview of the Turbidity Sensor

Our turbidity sensor, referred to as OpenProbe GLI-2, possesses two infrared LEDs and two infrared photodiodes in order to implement the GLI Method 2. For the infrared LEDs, we used the OSRAM SFH 4718A that has its peak wavelength at 860 nm, has a full width at half maximum (FWHM) of 34 nm, and supports up to 1000 mA of forward current. For photodetectors, we used the OSRAM SFH 2700 FA A01, a silicon PIN photodiode with a daylight-blocking filter that translates to a spectral sensitivity from 700 to 1100 nm.

The main functions required to use these optoelectronics components are LED drivers, transimpedance amplifiers, and an analog-to-digital Converter (ADC). Absorption underwater is stronger for longer wavelengths, so the use of IR photodiodes limits the influence of daylight during in situ measurements. However, due to the relatively small variations caused by turbidity, an ambient light-rejection strategy is still required and is taken care of by synchronous detection. While each of these functions can be achieved by discrete components, we chose to design our system around the ADPD1080 from Analog Devices, a highly integrated photometric front-end initially designed for photoplethysmography (PPG) in wearables or smartwatches, as it includes all the required features in a single low-power integrated circuit (IC), which is highly beneficial in terms of cost, miniaturization, and power consumption. Is possesses three LEDs drivers with up to 370 mA current capability, has the possibility to connect up to eight photodetectors to its transimpedance amplifier (TIA) with digitally adjustable gains and has an Analog Front-End (AFE) that is in charge of the rejection of signal offset and corruption due to the interference caused by ambient light and has a 14-bit ADC.

The block diagram in Figure 3 describes the overall architecture of the turbidimeter. An Adafruit Feather M0 microcontroller is used to control the different components, as it is a popular open-hardware configuration for environmental sensor projects [31,32]. Based around a low-power ATSAMD21G18 ARM cortex M0 processor, clocked at 48 MHz and 3.3 V logic, it can work with any 3.7 V Lithium polymer battery as power supply and integrates a charge circuitry. This microcontroller is available in different versions with wireless communication capabilities (BLE, LoRa, WiFi) with the same form factor, which allows to select the most appropriate communication standard based on the user needs. Depending on our needs, we used either the RFM95 LoRa version, which allows long-range wireless transmission of turbidity data, or the Bluefruit LE version (nRF51822 chipset for Bluetooth Low-Energy communications) that allows easy short-distance communication with a smartphone or a laptop, for example. Functionalities can easily be added in the form of add-on boards: for the data acquisition, we used the Adafruit Adalogger FeatherWing, which integrates a PCF8523 real-time clock and a microSD memory card socket to handle datalogging functions, i.e., timestamping and data recording as text files. The microcontroller controls the ADPD1080 photometric front-end through an I2C interface in order to adjust the various settings for LED drivers, TIA gain, and various timings.

### 2.2. Hardware Design

To achieve the communication between the ADPD1080 photometric front-end and the Adafruit Feather M0 microcontroller, additional components were required. An AP7313 low dropout voltage regulator was used to supply a clean 1.8 V voltage to the ADPD1080 from the Adafruit Feather M0 3.3 V output regulator. A PCA9306 I2C bus voltage-level translator was used between the 3.3 V logic level of the microcontroller and the 1.8 V logic level of the photometric front-end for the SDA and SCL lines, with 2.2 kohms pull-up resistors. Finally, an ADG3304 bidirectional logic level translator was used for the GPIO0 and GPIO1 pins, which are used for generating hardware interrupts on the microcontroller when data are available.

Optoelectronics components, i.e., LEDs and photodiodes, are integrated on a separate PCB to ensure proper positioning and implement the GLI Method 2. Due to the use of SMD components, the spatial distribution required, and the need of integration in a waterproof enclosure for in situ measurements, we chose to design a custom, flexible PCB that is bent in a circular shape to obtain proper positioning of the optical elements, as illustrated Figure 4A. Both PCBs are connected through Molex Picoblade 6 pins cable and connectors. The two-layer and the flexible PCBs are manufactured by the OSH Park company, and the components were assembled in-house using a reflow oven. The complete circuit diagram, CAD files, and pictures of the electronics are available in the repository given in Appendix A section (https://gitlab.laas.fr/vraimbau/OpenProbe (accessed on 2 January 2023)).

### 2.3. Sensor Housing

The literature is scant on GLI-2 ratio-based instruments for in situ turbidity measurement; based on the recent achievements presented in Table 1, we attribute this to the apparent complexity of building a waterproof enclosure for this four-beam design with equipment available in an academic facility. In order to make our design easily replicable, we tried to develop our sensor around off-the-shelf components and standard equipment/techniques that can be either outsourced or purchased. The waterproof enclosure is made by stereolithography (SLA) with a desktop Formlabs Form 3 3D printer and Black Resin, a methacrylate-based material. After development in isopropanol (Formlabs Form Wash), parts were cured overnight at 60 °C. This unusually long curing step is required for the subsequent overmolding step with polydimethylsiloxane (PDMS), as it has been observed that commercially available SLA resins inhibit its polymerization without this treatment [33]. The flexible PCB with its mounted LEDs and photodiodes was bent to be inserted within the enclosure with mechanical features that guide the LEDs and photodiodes to ensure proper alignment (Figure 4A).

We then used overmolding with PDMS (Sylgard 184—Dow Corning) to ensure waterproofing and optical transparency for the optical elements. This also ensures that no air was trapped in the housing, which is a key factor to obtain a sensor that can be used at depth in the water column. A 1:10 ratio PDMS mixture was poured over the 3D-printed enclosure and the flexible PCB with the help of a 3D-printed insert covered with a Kapton film in the center to create a smooth yet anti-sticking interface at each optical port. The whole assembly was polymerized at 65 °C overnight, and then, the insert with the Kapton film was removed, leaving optically clear and smooth windows in front of each optical element, as visible in Figure 4B,C with closeup views. In order to facilitate sensor testing and further replication, we designed and printed a M10 penetrator adapter to make our turbidity sensor compatible with the Blue Robotics waterproof enclosures that are regularly used in environmental sensor development [34]. We used a 2 in diameter, 100 mm long cast acrylic tube, which is rated for 300 m depth and houses the Adafruit Feather M0, the Adafruit Adalogger FeatherWing, our custom ADPD1080 PCB, and a LiPo battery as well as additional sensors for, in this case, pressure (depth) and temperature.

### 2.4. Software

The Adafruit Feather M0 is programmed through the Arduino IDE environment, with a custom library to handle the specific functionalities of the ADPD1080. The operation principle of the photometric front-end consists of the stimulation of the LEDs during short pulses (in our case, 3 µs duration) and the synchronous measurement of the returning signal from the photodetectors through the analog block. An integrator allows to sum up the returning signal from an adjustable number of pulses, allowing for an increase in the signal-to-noise ratio. The ADC output is obtained by the microcontroller either through the use of hardware interrupts (using GPIO0 and GPIO1 pins of the ADPD1080), which are generated each time new data are available in the ADC output register, or by data polling at a regular interval. While data polling is easier to implement, the use of hardware interrupt is more robust and allows for better efficiency of the code, especially if one considers optimizing the battery life of the system. To operate the GLI-2 method, four steps are required, as described Figure 5, which represents the measurement sequence as well as the timing diagrams. Settings of the photometric front-end for each measurement step were optimized. The pulse number per step is 50, the TIA gain is set to 50 kΩ, and LED current is set to 260 mA for active measurement (nephelometric) and 70 mA for reference measurement (attenuation). The AFE possesses an internal averaging function that allows to lower the noise at the expense of a longer response time and higher power consumption. We set the averaging factor to 8, which using the optimized settings lead to a response time of about 100 milliseconds for a complete measurement cycle. As a comparison, our handheld AQ3010 device takes approximately 20 s to deliver a measurement. This very short response time is particularly valuable to measure turbidity profiles, i.e., variation of turbidity versus depth. Increasing the averaging factor above 8 only resulted in a relatively small improvement on the noise level.

### 2.5. Formazin Standard Calibration Method

While the calibration of turbidimeters with Formazin suffers from limitations, as mentioned in the introduction section, this calibration method is the current standard of reference methods, and as such, it was selected in this work. Turbidity calibration solutions were made by dilution of a 4000 NTU Formazin Turbidity Standard (Hach) in laboratory-grade deionized water. Solutions were prepared daily to avoid stability issues and were remixed prior to measurements to prevent the suspension from settling out. Solutions were then measured using our reference instrument, a commercial Thermo Scientific Aquafast AQ3010 Turbidity Meter, which is a handheld device that uses 90° nephelometric method and outputs turbidity in NTU units. Figure 6 summarize the data obtained with this instrument for 0 to 50 NTU solutions. The increments between each solution were adapted to the turbidity levels, with 0.5 NTU increments in the low range and 10 NTU increments in the high range. The 0 NTU blank solution is the same laboratory-grade deionized water used for dilutions of the 4000 NTU Formazin standard.

## 3. Results

### 3.1. Photodetector Current

The ADPD1080 photometric front-end is a complex component with many different settings that can influence drastically its performance. Prior to its use, we chose to validate the behaviour of our optoelectronic component selection and enclosure design through an experiment using benchtop instruments to stimulate LEDs and measure photodiodes currents exposed to a range of turbidity calibration solutions in controlled laboratory conditions (i.e., no variations in ambient light).

Briefly, a Keithley 2400 source meter was used to stimulate the LEDs with a constant current of 80 mA, while the photocurrent issued from the photodiodes was measured with a Keithley 2100 Multimeter set up as an ampere meter. The LEDs excitation current was only briefly maintained during the measurement to avoid detrimental heating effects. The two optical configurations required by the GLI-2 method, i.e., 90° nephelometric (referred also as active) and 180° attenuation (referred also as reference), were measured with this setup and shown in Figure 7 for turbidity solutions varying from 0 to 40 NTU. It can be noted that the photocurrents vary as expected: in 90° nephelometric configuration, an increase in turbidity results in an increase of light diffraction and, consequently, in an increase of the collected light by the active photodiode. In the attenuation configuration, an increase of turbidity leads to an increase of light scattering and absorption, which turns into a decrease of the collected light by the 180° photodiode. A linear relationship between photocurrent and turbidity was observed in all configurations.

For both active and reference optical configuration, slight discrepancies can be observed, which could be attributed to individual optoelectronic components differences or optical effects due to misalignment or differences in the PDMS transparency. It should be emphasized that the photocurrent variations are rather small and correspond to approximately 10 nA per NTU in 90° nephelometric configuration and 40 nA per NTU in 180° attenuation configuration. Thus, the corresponding photocurrent variation to a 0.1 NTU turbidity variation shall be in the range of a nA. Nonetheless, these tests confirm that our sensor design works as expected in the 0 to 40 NTU range.

### 3.2. Sensor Calibration

After this sensor design validation using benchtop instruments, we then replaced the benchtop source meter and the ampere meter with our custom PCB hosting the ADPD1080 photometric front-end and its additional components. In order to implement ambient light rejection, the excitation light was now modulated and consisted of trains of short, 3 µs pulses, while the scattered and/or absorbed resultant signal was synchronously sampled. Photocurrents generated by the photodiodes were internally amplified by a transimpedance amplifier and conditioned prior to being converted by a 14-bit ADC, giving an output in counts. In order to translate these counts to turbidity-related units, a calibration had to be performed for both configurations, namely 90° nephelometric and 180° attenuation. The optimized settings for each configuration are given in the Software section. Formazin turbidity calibration samples were prepared, covering a range from 0 to 50 NTU, with 0.5 NTU increments in the 0 to 10 NTU range, and 10 NTU increments otherwise.

Figure 8 shows the obtained results in both optical configurations for Active1, Reference1, Active2, and Reference2 signals. It can be observed that each channel has slightly different characteristics in terms of offset and sensitivity; however, both exhibit similar tendencies. In the nephelometric configuration, sensitivity varies approximately from 120 (Active1) to 140 (Active2) counts per NTU, while in the attenuation configuration, sensitivity varies approximately from 70 (Reference1) to 80 (Reference2) counts per NTU. However, these differences are not considered as a major issue thanks to the ratiometric nature of the GLI-2 method: in our case, a slightly lower sensitivity is observed on Active1 and Reference1, which means that the photodetector PD1 generates a lower photocurrent than PD2 in the same conditions. As Active1 is in the numerator of Equation (2) and Reference1 in the denominator, this difference is cancelled out. This is the same mechanism that gives the GLI-2 some advantages toward biofouling, as if a biofilm partially obstructs an optical port, the sensitivity decrease will be cancelled out by the aforementioned principle. As expected, the 90° nephelometric configuration is more precise and sensitive for the low-turbidity range, i.e., 0 to 10 NTU, as the variation in absorption in this range is very small.

A three-point calibration was performed, as recommended by the U.S. Geological Survey for submersible turbidity sensors [35]. The sensor was immersed in three Formazin calibration solutions of 0, 10, and 40 NTU, while the ADC counts for Active1, Reference1, Active2, and Reference2 signals were recorded. The raw GLI-2 output was calculated according to Equation (1) and plotted Figure 9.
(1)$${\mathrm{GLI-}2}_{\mathrm{raw}}=\sqrt{\frac{Active1*Active2}{Reference1*Reference2}},$$

From this calibration curve, the calibration coefficients *Cal_slope_* and *Cal*_0_ were calculated from the linear regression fit of the three-point calibration curve shown Figure 9 to satisfy following equation:(2)$${\mathrm{GLI-}2}_{\mathrm{NTU}}=Cal_{slope}\sqrt{\frac{Active1*Active2}{Reference1*Reference2}}-Cal_{0},$$

With the optimized settings described in the Software section, the final calibration equation corresponds to the values below:(3)$${\mathrm{GLI-}2}_{\mathrm{NTU}}=285.714\sqrt{\frac{Active1*Active2}{Reference1*Reference2}}-110.257,$$

These calibration coefficients were then used to update corresponding variables in the microcontroller code, so the sensor was able to directly output turbidity values in NTU units. Figure 10 shows the results obtained in the 0 to 10 NTU range, with 0.5 NTU increments, and in the extended range of 0 to 50 NTU. Our calibrated GLI-2 sensor data were plotted together with a confidence interval of +/−0.4 NTU around the ideal value, highlighting the good fidelity of the sensor even at these low turbidity values.

While we focused on the 0 to 50 NTU range, good linearity was observed up to 100 NTU, but due to the large volumes of calibration standard required to fully immerse our sensor, we chose to focus on the lower turbidity range. While we did not perform any testing above 100 NTU, the sensor should also work at higher turbidity values to the extent that the photometric front-end settings and the calibration curve are optimized for this range, as similar configurations have been successfully used up to 1000 NTU. We finally took the opportunity to compare our sensor implementing the GLI-2 method to the commercial Thermo Fisher Aquafast AQ3010 instrument, the portable handheld device used during our experiments to assess the quality of our Formazin calibration dilutions that cost approximately USD 1000. The intercomparison plot is given Figure 11.

The data show that our sensor compares nicely even in the 0 to 10 NTU range despite an overall BoM cost of approximately USD 50 for a single prototype (excluding the Blue Robotics high-pressure enclosure, which could be replaced by a home-made PVC-based enclosure to keep the costs down), with an accuracy of +/−0.4 NTU or better in the 0 to 10 NTU range. While the commercial AQ3010 offers better accuracy, it is not capable of in situ measurement, as it requires manual water sampling, followed by pipetting of the sample into a clean vial, as well as a 20 s response time compared to the 100 ms response time of our OpenProbe GLI-2 sensor. In terms of cost, the Hydrolab 4-beam turbidity sensor, one of the very few commercial sensors capable of implementing the GLI-2 method in situ, costs several thousands of dollars.

## 4. Discussion

It is well known that measuring low turbidity values is particularly challenging, especially if one considers the additional constraints of an in situ deployable instrument, as this adds some complexity in the design to make it fully submersible and some additional issues to handle such as ambient light variation, biofouling, or temperature variations. Low-cost turbidity sensor development is an active research field, as turbidity is a ubiquitous indicator of water quality, and as such, it is a parameter of interest in many fields from academic research to water agencies or recreational activities such as swimming. While many recent studies have shown great developments (as summarized in Table 1), there currently seems to be no low-cost solution for in situ measurement in the low-turbidity range. In this project, we thus developed a prototype of a low-cost turbidity meter that is capable of measuring turbidity values in the range of 0 to 50 NTU with an accuracy of +/−0.4 NTU after calibration. Compared to a commercial portable handheld instrument, our sensor shows comparable performance at a fraction of the cost. Furthermore, by using a design based on the GLI-2 method, integrating an ambient light-rejection strategy using an integrated photometric front-end, and developing a simple yet effective waterproof enclosure based on SLA 3D printing and PDMS overmolding, this sensor can be deployed in situ in natural waters, as the GLI-2 method offers inherent robustness toward biofouling, LED and photodiode drifts or color effects. Our future works will focus on long-term field validation of our sensor in water bodies exposed to significant turbidity variations as well as an intercomparison campaign with a commercial, in situ GLI-2 sensor in order to further validate these encouraging results.

## Figures and Tables

**Figure 1 sensors-23-01897-f001:**
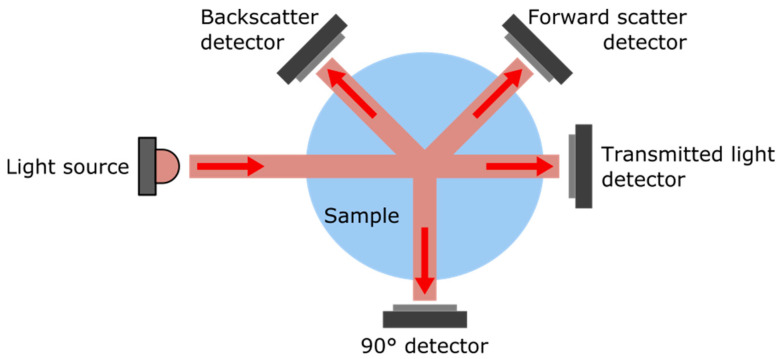
Illustration of the most common optical configurations adopted in turbidity measurement devices.

**Figure 2 sensors-23-01897-f002:**
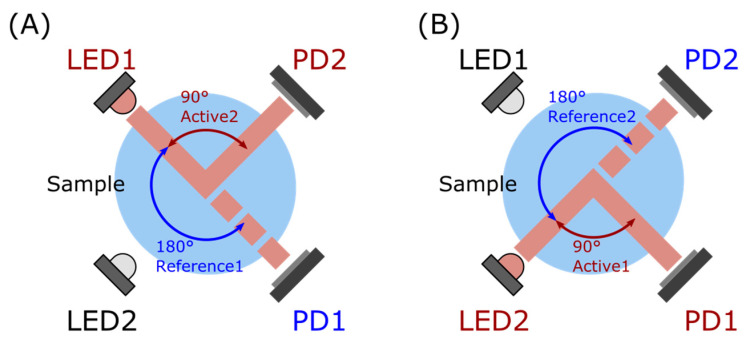
Illustration of the GLI-2 method, a ratiometric method based on a modulated 4-beam design. (**A**) Phase one: light source LED1 is on, photodetector PD2 measures the Active2 signal (90° nephelometric), and photodetector PD1 the Reference1 signal (180° attenuation). (**B**) Phase two: light source LED2 is on, photodetector PD2 is the Reference2 signal (180° attenuation), and photodetector PD1 is the Active1 signal (90° nephelometric).

**Figure 3 sensors-23-01897-f003:**
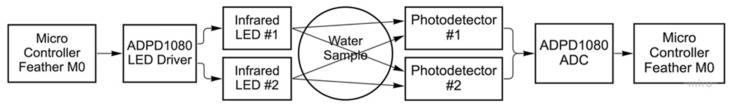
Block diagram of the OpenProbe GLI-2 sensor.

**Figure 4 sensors-23-01897-f004:**
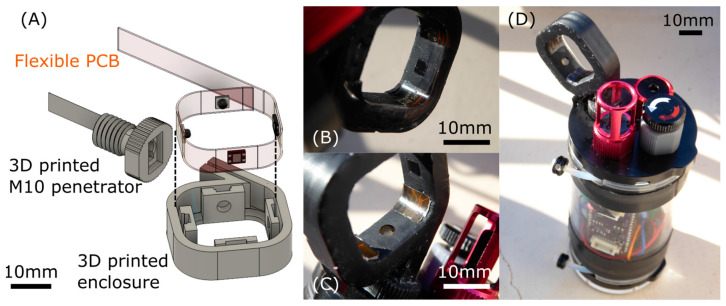
(**A**) CAD illustration of the GLI-2 sensor, with the flexible PCB hosting the two IR LEDs and the two photodiodes, the 3D-printed enclosure, and a 3D-printed M10 penetrator that makes the sensor compatible with Blue Robotics waterproof enclosures. (**B**,**C**) Close-up pictures of a photodiode and an LED optical port, respectively, after the PMDS overmolding step, to illustrate the good transparency and optical properties obtained with our method. (**D**) Implementation for in situ deployment with the electronics and LiPo battery protected behind a Blue Robotics two-inch diameter enclosure, showing the GLI-2 sensor head as well as additional pressure (depth) and temperature sensors.

**Figure 5 sensors-23-01897-f005:**
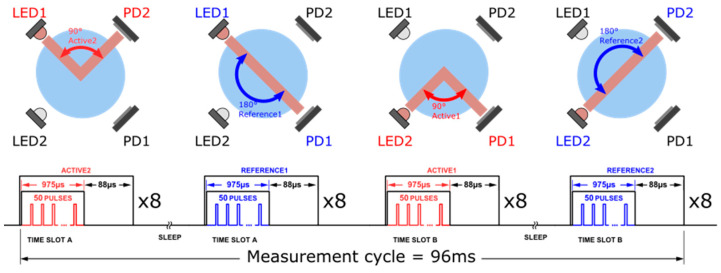
Measurement sequence implemented by the software to measure Active2, Reference1, Active1, and Reference2 signals and corresponding timing diagram illustrating the AFE operation. Each step was performed eight times to perform internal averaging, which allows to improve signal-to-noise ratio. Total measurement time in this configuration is 96 ms.

**Figure 6 sensors-23-01897-f006:**
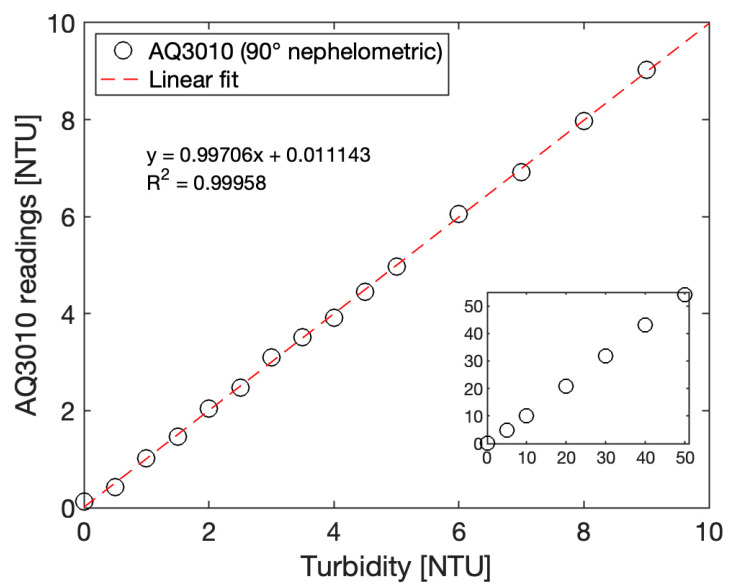
Graph of a calibrated Thermo Scientific Aquafast AQ3010 Turbidity meter to Formazin solutions from 0 to 10 NTU (main graph) and 0 to 50 NTU (inset).

**Figure 7 sensors-23-01897-f007:**
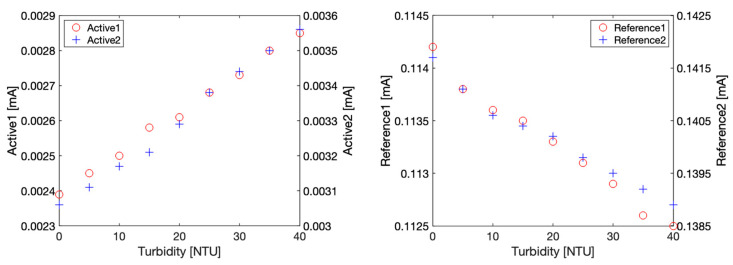
Photodetector current characterization obtained with benchtop instruments with Formazin solutions ranging from 0 to 40 NTU. **Left***:* 90° nephelometric configuration current for Active1 and Active2 signals. **Right***:* 180° attenuation configuration current for Reference1 and Reference2 signals.

**Figure 8 sensors-23-01897-f008:**
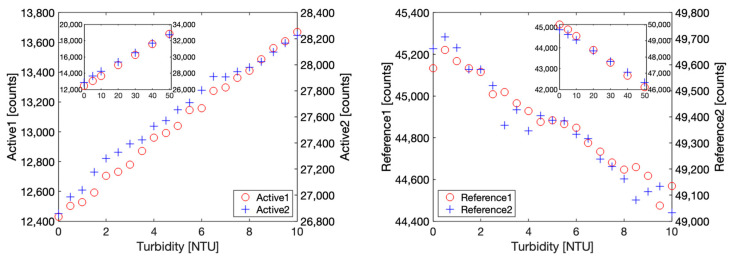
Calibration experiment with Formazin solutions showing ADC output expressed in counts. Main graphs represent data from 0 to 10 NTU with 0.5 NTU increments, while inset shows the 0 to 50 NTU range. **Left***:* 90° nephelometric configuration ADC counts for Active1 and Active2 signals. **Right***:* 180°attenuation configuration ADC counts for Reference1 and Reference2 signals.

**Figure 9 sensors-23-01897-f009:**
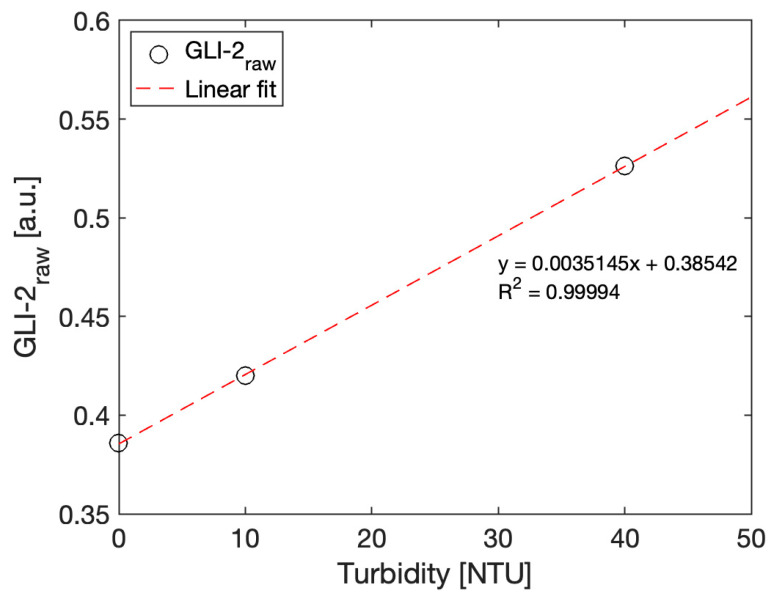
Three-point calibration of the sensor. GLI-2_raw_ values are calculated from the Active1, Active2, Reference1, and Reference2 signals according to Equation (1) against three calibration solutions of 0 (deionized water), 10, and 40 NTU (Formazin dilutions).

**Figure 10 sensors-23-01897-f010:**
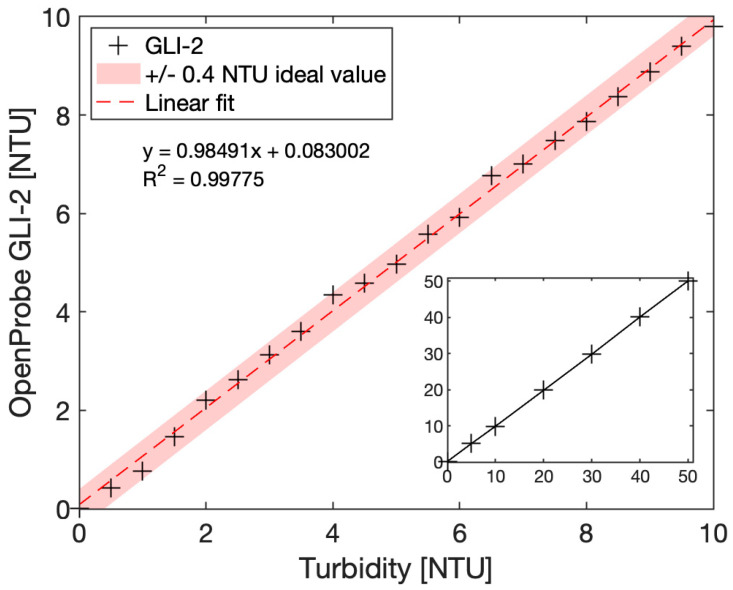
Calibrated OpenProbe GLI-2 sensor immersed in Formazin calibration solutions from 0 to 10 NTU (main graph) and 0 to 50 NTU (inset). A confidence interval of +/−0.4 NTU was obtained in the 0 to 10 NTU range.

**Figure 11 sensors-23-01897-f011:**
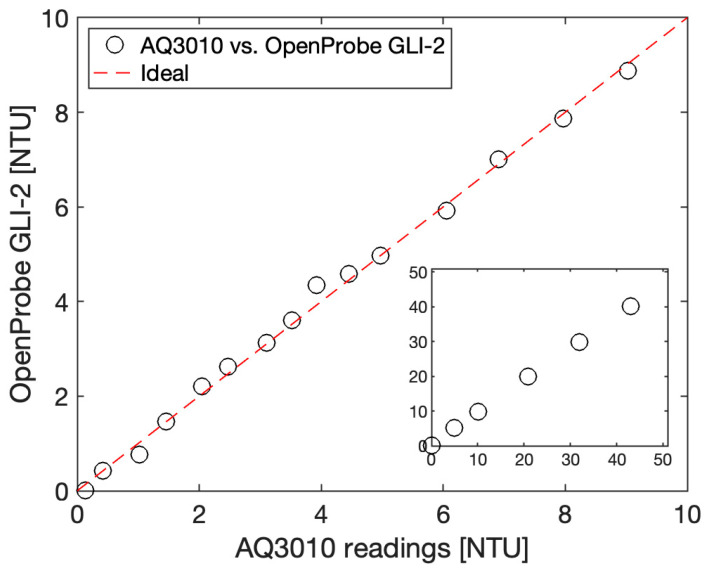
Intercomparison of our turbidity sensor, OpenProbe GLI-2, versus a portable handheld Thermo Fisher AQ3010.

**Table 1 sensors-23-01897-t001:** Recent achievements in the literature on turbidimeter developments.

Sensor	Range	Resolution	In Situ	Method	Reference
Fay et al.	0–100 NTU 0–1000 NTU	N.A.	No	ISO 7027	[24]
Kitchener et al.	N.A.	N.A.	No	TARDIIS	[22]
Gillett et al.	0–100 NTU	1 NTU	No (continuous)	Nephelometry	[25]
Trevathan et al.	100–400 NTU		Yes	Attenuation	[3]
Zang et al.	40–300 NTU	3 NTU	No	Nephelometry and attenuation	[26]
Matos et al.	0–4000 NTU	N.A.	Yes	IR backscatter, nephelometry and attenuation	[2]
Metzger et al.	0.1–1000 NTU	0.04 to 3 NTU	No	ISO 7027	[27]
Parra et al.	0–200 NTU	N.A.	No	Attenuation	[28]
Kelley et al.	0–1000 NTU	0.02 NTU	No	Nephelometry	[29]
Our work	0–100 NTU	0.4 NTU	Yes	GLI-2	N.A.

## Data Availability

The data presented in this study are available in https://gitlab.laas.fr/vraimbau/OpenProbe (accessed on 2 January 2023).

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
