# Peer review of "Development of a Frugal, In Situ Sensor Implementing a Ratiometric Method for Continuous Monitoring of Turbidity in Natural Waters"

_sensors, 2023, doi:10.3390/s23041897_

Round 1
Reviewer 1 Report
The manuscript titled “Development of a frugal, in-situ sensor implementing a ratiometric method for continuous monitoring of turbidity in natural waters” presented detailed hardware and software to build a GLI 2 based turbidimeter that can operate continuously in-situ in natural water at low cost. The proposed sensor was tested against a commercial benchtop turbidimeter with Formazin standards, and a range of 0 to 40 NTU, with an average error of +/- 5% was demonstrated. In result, this manuscript provided a low-cost solution for in-situ measurement in the low turbidity range. I suggest the following issues should be addressed before the manuscript is accepted.
1. Since GLI Method is known to be very accurate for lower turbidity ranges in particular within the 0-1 NTU range, and the target of this work is the coastal waters with a turbidity range from 0 to 10 NTU typically during the year. However, the sensor was tested with Formazin solutions ranging from 0 to 40 NTU in 5 NTU increments. Could more data in lower turbidity range be provided?like 0-10 NTU in 1 NTU increments?
2. In Table 1, Range and Resolution were given to evaluate different type of turbidimeters, but only range was given in the test, Could the resolution of the proposed sensor was measured?
Author Response
1.
We agree with this comment and have modified the manuscript accordingly, with a main focus on results obtained in the 0 to 10 NTU range, with 0.5 NTU increments.
We also took the opportunity to update the manuscript with our latest experimental results, that have been obtained with an updated reference for the photodiodes that resulted in a significant improvement in the overall performance. The results from figure 7 to Figure 11 have been updated with these latest results, and the section “Material and Methods” have been updated with the new photodiode reference.
2.
In the range of 0 to 10 NTU, the sensor resolution (with the updated photodiodes and the latest results presented in the revised manuscript) is estimated to 0.4 NTU. This resolution can be enhanced by increasing the averaging factor setting of the photometric front-end, which is currently set to 8, to improve the Signal to Noise Ratio, however it has a direct (proportional) impact on the response time.
The manuscript has been modified to indicate the sensor resolution, and a comment on the response time has been added to the 2.4. Software section.
Reviewer 2 Report
Turbidity is an important indicator of water quality in rivers, streams, lakes, sea and watershed, and measuring low turbidity values is particularly challenging, especially the in-situ deployable instrument. This project proposed an open-source, low-cost in-situ turbidity sensor with a suitable sensitivity and operating range to operate in low to medium turbid natural waters. The sensor was tested in laboratory conditions against a commercial benchtop turbidimeter with Formazin standards, and in field conditions. The monitoring results were analyzed getting a linear trendline from 0 to 40 Nephelometric Turbidity Unit (NTU). The project provides a new academic work on a GLI 2 based design that can operate turbidity continuously in-situ. The paper can be accepted after minor modification.
1. Figure 1 is too simple, the particles in the picture are not very clear.
2. The format of Figure 2 to Figure 5 should be consistent, including the color of the “Light Beam”, the form of the arrow, and the icon of the detector.
3. Some expressions in the paper are repeated, such as lines 274-275 and lines 351-352. The text needs to be checked and revised.
4. Section 2.5, the solution at 0 NTU turbidity also needs to be tested.
5. In Figure 11, the thickness of each vertical axis and its corresponding title should be the same.
6. In lines 472-473, how does the ratiometric nature of the GLI-2 method cancel out the differences caused by different sensitivity of the nephelometric and attenuation configuration?
7. Please verify the “XX” in lines 481, 490 and 500 refers to which figure or equation.
8. Is the GLI-2 value in Figure 13 and Figure 14 adopted the same calculation method? If so, why only 3 points are used for correction in Figure 13, and why there is a big difference between the fitting formula in Figure 14 and Equation 3?
9. In lines 507-508, with whom are the percentage errors of the two sensors compared? How is it calculated?
10. The paper is helpful for the authors to understand turbidity more vividly. A methodology to predict the run-out distance of submarine landslides. Computers and Geotechnics, 153, 105073.
Author Response
1.
As requested by reviewer 3, the introduction has been reduced, and Figure 1 has been removed.
2.
For the same reasons as above Figures 2 to 4 have been replaced by a single figure (now Figure 1) that illustrates the most common optical configurations for turbidity measurements. Figure 5 (now Figure 2) has been slightly modified as the indices used for the signals and the photodiodes were not consistent with the typical denomination found in the literature.
3.
The repetition of these expression has been addressed.
4.
As the 0 NTU is only lab grade deionized water and not a dilution of the 4000 NTU Formazin standard, it was mentioned separately in the text. However the 0 NTU solution is also tested and presented in the Figure 10 (now Figure 6) graph. The text has been modified to avoid the confusion.
5.
All figures have been updated and standardized, with an emphasis on the low turbidity range to better suit the objectives of the article.
6.
The difference in sensitivity between each channel (i.e. between Active1 and Active2 for nephelometric, and between Reference1 and Reference2 for attenuation) is generally due to a difference in the optical system, for example a default in the optical window in front of a LED or a photodiode. Let say that the photodiode PD1 photocurrent is 10% lower than the PD2 photocurrent in the same conditions: this will result in a decreased sensitivity that will affect all measurements that uses PD1, in that case Active1 and Reference1 signals. As Active1 is in the numerator of equation (2), and Reference1 in the denominator, this 10% difference is cancelled out.
This is the same mechanism that gives the GLI-2 some advantages toward biofouling, as if a biofilm partially obstructs one optical port, the sensitivity decrease will be cancelled out by the aforementioned principle.
These explanations have been added to the manuscript
7.
We apologize for these errors. The authors have corrected the manuscript to replace XX by the appropriate index numbers.
8.
Figure 13 (now Figure 9) represents the calibration of the sensor, and plots the GLI2_raw values which are calculated directly from the ADC counts (values shown on Figure 12 (now Figure 8) for all channels), according to Equation (1). At this stage, GLI2_raw is dimensionless and the sensor cannot provide readings in NTU. Then a linear regression is done on the (GLI2_raw vs. turbidity) curve, to obtain the calibration coefficients Cal_slope and Cal_0, which are introduced into equation (2) to get equation (3). Once the calibration is performed the sensor can deliver an output expressed in NTU.
There was an error in the coefficients of Equation (3), which was not displaying the Cal_slope and Cal_0 values but the slope and offset value of the three-point calibration curve of Figure 13 (now Figure 9). The information has been corrected.
From the linear fit coefficients values displayed Figure 13 (now Figure 9):
〖Cal〗_slope=1/0.0035145=285.74
And
〖Cal〗_0=0.385421/0.0035145=110.257
Regarding the three-point calibration, it has been selected as it is the recommended technique for the calibration of submersible turbidity sensors, as mentioned in the following reference p.29-30: Anderson, C.W., 2005, Turbidity: U.S. Geological Survey Techniques of Water-Resources Investigations, book 9, chap. A6.7, https://doi.org/10.3133/twri09A6.7.
The GLI-2 method is known to be linear in the 0 to 100 NTU (with greater precision in the 0 to 40NTU range), so the relationship between GLI2 output and turbidity is obtained with a simple linear regression. Preparation of calibration solutions below 10 NTU is more prone to errors, so we choose to use 10 and 40 NTU calibration standards and laboratory grade deionized water for 0 NTU.
The manuscript has been completed with the USGS reference to justify the three-point calibration method.
9.
The percentage error is calculated directly as the difference between the ideal value of the turbidity calibration standard and the response delivered by the sensor. The percentage of error given in the manuscript for the commercial AQ3010 and our GLI2 sensor is the worse value observed over the measured range. However, as this percentage is not easy to interpret (as it can reach pretty high values in very low turbidity), we updated the manuscript to emphasis one the accuracy of the sensor as we believe it is more relatable.
10.
We agree with this comment, as the measurement of turbidity currents is also of great interest. Two references have been added to the introduction.
Reviewer 3 Report
The manuscript “Development of a frugal, in-situ sensor implementing a ratiometric method for continuous monitoring of turbidity in natural waters” by Raul Sanchez et al. reported a sensor for turbidity monitor in natural waters. There are some comments for authors to improve their manuscript:
1. It is an article research, but authors spend 7 pages for introduction, which makes the manuscript looks like a review paper. The description and some similar figures in this part should be brief.
2. Some unnecessary figures such as Fig. 7 and 8 can be moved to the appendix.
3. The detail dimensions of the sensor can be mark in Fig. 9.
4. Sec. 2.4 gives less information for readers, it can be give the flow chart of the whole system better than the signal control of the pins.
5. It seems an incomplete manuscript, such as Line 481, 490, 500, “equation XX” and “figure XX”.
6. Performance comparison of the designed sensors and others should be listed as Table. 1.
7. Why is only 3 points in Fig. 13 for calibration?
8. Authors said that Fig. 15 shows the comparison of designed and a commercial sensor, however, it cannot find two different data in this figure.
Author Response
1.
We agree with this comment. The introduction has been thoroughly modified to be more straightforward. In particular, information from Figure 2 to 4 have been compiled on a single Figure 1 to illustrate the most common configurations for turbidity measurements, and informations about the various method have been greatly simplified, pointing to appropriate reference in the literature.
2.
As requested per the reviewer, the pictures of Figure 7 and Figure 8 have been moved to the supplementary materials (Direct link to the pictures : https://gitlab.laas.fr/vraimbau/OpenProbe/-/tree/main/Turbidity/Pictures).
3.
Scale bars have been added to all the pictures of Figure 9 (now Figure 4). More detailed dimensions can be obtained from the mechanical files in the supplementary information.
4.
Table 2 has been replaced by a combination of a chart showing the different steps of a measurement sequence, enriched with a timing diagram to illustrate the operating mode of our sensor. Main settings (counts, TIA gain, LED current) have been added to the main text in order to keep this information easily accessible to the reader. Some information about the sensor’s response time have also been added.
5.
We apologize for this. The manuscript has been updated with the appropriate equations and figures numbers.
6.
Table 1 has been updated with the performance of our sensor.
7.
Three-point calibration has been selected as it is the recommended technique for the calibration of submersible turbidity sensors, as mentioned in the following reference p.29-30 : Anderson, C.W., 2005, Turbidity: U.S. Geological Survey Techniques of Water-Resources Investigations, book 9, chap. A6.7, https://doi.org/10.3133/twri09A6.7.
The GLI-2 method is known to be linear in the 0 to 100 NTU (with greater precision in the 0 to 40NTU range), so the relationship between GLI2 output and turbidity is obtained with a simple linear regression. Preparation of calibration solutions below 10 NTU is more prone to errors, so we choose to use 10 and 40 NTU calibration standards and laboratory grade deionized water for 0 NTU.
The manuscript has been completed with the USGS reference to justify the three-point calibration method.
8.
Figure 15 (now Figure 11) is a plot of the Aquafast AQ3010 turbidimeter data in x axis (commercial sensor), versus calibrated GLI2 output data (our sensor) in y axis.
The authors have chosen this type of representation as it is commonly used for intercomparison between two different sensors, and give a direct view on the correlation between the two. If both sensors are plotted on the same graph against turbidity, the data points are almost superimposed and thus we found the resulting graph was harder to interpret for the reader.
Round 2
Reviewer 3 Report
Authors have revised the manuscript as reviewers' comments.